# Optimal Combination of Mother Wavelet and AI Model for Precise Classification of Pediatric Electroretinogram Signals

**DOI:** 10.3390/s23135813

**Published:** 2023-06-22

**Authors:** Mikhail Kulyabin, Aleksei Zhdanov, Anton Dolganov, Andreas Maier

**Affiliations:** 1Pattern Recognition Lab, University of Erlangen-Nuremberg, 91058 Erlangen, Germany; andreas.maier@fau.de; 2Engineering School of Information Technologies, Telecommunications and Control Systems, Ural Federal University Named after the First President of Russia B. N. Yeltsin, Yekaterinburg 620002, Russia; a.e.zhdanov@urfu.ru (A.Z.); anton.dolganov@urfu.ru (A.D.)

**Keywords:** biomedical research, electroretinography, electroretinogram, ERG, classification, deep learning, cnn, transformer, wavelet, scalogram

## Abstract

The continuous advancements in healthcare technology have empowered the discovery, diagnosis, and prediction of diseases, revolutionizing the field. Artificial intelligence (AI) is expected to play a pivotal role in achieving the goals of precision medicine, particularly in disease prevention, detection, and personalized treatment. This study aims to determine the optimal combination of the mother wavelet and AI model for the analysis of pediatric electroretinogram (ERG) signals. The dataset, consisting of signals and corresponding diagnoses, undergoes Continuous Wavelet Transform (CWT) using commonly used wavelets to obtain a time-frequency representation. Wavelet images were used for the training of five widely used deep learning models: VGG-11, ResNet-50, DensNet-121, ResNext-50, and Vision Transformer, to evaluate their accuracy in classifying healthy and unhealthy patients. The findings demonstrate that the combination of Ricker Wavelet and Vision Transformer consistently yields the highest median accuracy values for ERG analysis, as evidenced by the upper and lower quartile values. The median balanced accuracy of the obtained combination of the three considered types of ERG signals in the article are 0.83, 0.85, and 0.88. However, other wavelet types also achieved high accuracy levels, indicating the importance of carefully selecting the mother wavelet for accurate classification. The study provides valuable insights into the effectiveness of different combinations of wavelets and models in classifying ERG wavelet scalograms.

## 1. Introduction

The pediatric electroretinogram (ERG) is a measure of the electrical activity of the retina in response to light stimulation, typically performed on infants and children. The ERG signal consists of a series of positive and negative waveforms, labeled as a-wave, b-wave, and c-wave, which reflect the activity of different retinal cells [1]. The a-wave represents the photoreceptor response, while the b-wave reflects the activity of the bipolar cells and Müller cells [2]. The amplitude and latency of the a-wave and b-wave are commonly used as parameters to evaluate the function of the retina in pediatric patients. Abnormalities in the pediatric ERG signal can be indicative of a range of retinal diseases or disorders, including congenital stationary night blindness, retinitis pigmentosa, and Leber’s congenital amaurosis.

The c-wave of the ERG is often considered less prominent and less studied compared to the a-wave and b-wave. The a-wave represents the initial negative deflection, reflecting the hyperpolarization of photoreceptors, while the b-wave represents the subsequent positive deflection, primarily originating from bipolar cells [3]. These waves are well-established and have been extensively researched due to their direct relevance to the visual pathway.

In contrast, the c-wave represents a slower, positive deflection following the b-wave, originating from the retinal pigment epithelium (RPE) and Müller cells [4]. Its amplitude is smaller and its clinical significance is not as clearly understood [5]. Consequently, it receives less attention in scientific literature and clinical practice.

While the a-wave and b-wave are directly related to photoreceptor and bipolar cell function [6], the c-wave is thought to reflect RPE and Müller cell activity, which are involved in the retinal pigment epithelium-photoreceptor complex. The complex nature of these cells and their functions may contribute to the relatively limited emphasis on the c-wave in the text compared to the a-wave and b-wave. However, further research is needed to fully understand the c-wave’s role and its potential clinical applications.

ERG may be of various types, depending on the specific electrophysiological protocol used. One type of ERG signal is the Scotopic 2.0 ERG response, which is obtained under conditions of low light intensity. This response is mainly generated by the rod photoreceptor cells in the retina and is characterized by a relatively slow and low-amplitude waveform. Another type of ERG signal is the Maximum 2.0 ERG response, which represents the maximum electrical response that can be elicited from the retina. This response is obtained under conditions of high light intensity and is mainly generated by the cone photoreceptor cells in the retina. The Maximum 2.0 ERG response is characterized by a faster and higher-amplitude waveform compared to the Scotopic 2.0 ERG response. The Photopic 2.0 ERG response is a third type of ERG signal that is obtained under conditions of moderate light intensity. This response is also mainly generated by the cone photoreceptor cells in the retina and is characterized by a waveform that is intermediate in both amplitude and latency between the Scotopic 2.0 ERG response and the Maximum 2.0 ERG response. It is important to note that the specific electrophysiological protocol used to obtain these different types of ERG signals can vary depending on the research question and the clinical application. Detailed information about the parameters of the electrophysiological study, including the light intensity, wavelength, and duration of the light stimulus, as well as the recording electrodes and amplification settings are shown in a previous study [7].

The Figure 1 depicts pediatric ERG signals of both healthy and unhealthy subjects, along with the designation of the parameters that clinicians analyze. By analyzing the parameters of the ERG waveform, such as the amplitude (a, b) and latency of the a-wave and b-wave (la, lb), clinicians can identify abnormalities and diagnose a range of retinal disorders [8].

In a healthy subject in Figure 1a, the temporal representation of the ERG signal typically exhibits distinct and recognizable waveforms. The signal begins with a negative deflection called the a-wave, which represents the hyperpolarization of photoreceptors in response to light stimulation [1]. Following the a-wave, there is a positive deflection known as the b-wave, which primarily reflects the activity of bipolar cells in the retina. The b-wave is usually larger in amplitude compared to the a-wave [9].

In an unhealthy subject in Figure 1b, the shape of the ERG signal in temporal representation can vary depending on the underlying pathology [10]. In some cases, there may be a significant reduction or absence of both the a-wave and b-wave, indicating a severe dysfunction or loss of photoreceptor and bipolar cell activity [11]. This can be observed in conditions such as advanced retinitis pigmentosa or severe macular degeneration.

Alternatively, certain diseases may selectively affect specific components of the ERG waveform. For example, in some cases of congenital stationary night blindness, the b-wave may be reduced or absent while the a-wave remains relatively normal, indicating a specific defect in bipolar cell function [12,13].

Thus, the shape of the ERG signal in temporal representation provides valuable insights into the integrity and function of retinal cells and can aid in the diagnosis and understanding of various retinal diseases and disorders [14].

In this study, we implemented trained deep-learning models using images of wavelet scalograms to determine the optimal mother wavelet. This approach differs from previous works, where the authors independently searched for features from the time-frequency representation of the signal. It should be noted that deep learning is optimal for image classification because it allows for the extraction of high-level features from raw data, which can result in higher accuracy. While adult ERGs can be standardized by establishing norms for various parameters, pediatric ERGs are less specific as the amplitude and latency of such ERGs can vary considerably. Consequently, diagnosis often necessitates the use of supplementary diagnostic methods. The study’s scientific novelty resides in utilizing deep learning techniques to identify optimal mother wavelet for pediatric ERGs, an approach that can also be applied to other types of ERG signals.

The Section 2 explores prior studies using diverse wavelets to analyze adult ERG data and a recent study utilizing deep learning to identify the optimal mother wavelet for pediatric ERGs. In the Section 3, we describe an approach to address ERGs class imbalance through under-sampling the majority class, applying wavelet transformation, and training five deep learning models for ERGs classification. The Section 4 presents findings from the experiment, demonstrating the highest median accuracy with the Ricker Wavelet and Vision Transformer combination for ERG wavelet scalogram classification. The Section 5 addresses limitations and emphasizes the significance of expanding the feature space via continuous wavelet transform for effective classification. Finally, the Section 6 highlight the efficacy of the combination of Ricker Wavelet and Vision Transformer in achieving high accuracy for ERG wavelet scalogram classification.

## 2. Related Works

Wavelet analysis has been widely used to study ERG in the field of ophthalmology. In recent publications shown in Table 1, the selection of the mother wavelet has been motivated by various factors. In the study presented in [15], mother wavelets were optimized to analyze normal adults’ ERG waveforms by minimizing scatter in the results. This approach led to improved accuracy and allowed for a more precise analysis of the data.

Different wavelets emphasize various features of a signal, making it crucial to choose the most appropriate mother wavelet. In a previous study [16], researchers conducted a preliminary analysis and concluded that the Ricker wavelet was the best fit for their waveforms due to its conformity to the shape of the adult ERG data. Similarly, another study [17] suggested that the Morlet wavelet was appropriate for adult ERG analysis, although there is still no consensus on the optimal mother wavelet. The aforementioned articles successfully addressed the classification problem and provided frequency pattern estimates for ERG.

The use of the Morlet wavelet transform in ERG analysis has been shown to provide a more comprehensive analysis of the data. For example, in [18], the Morlet wavelet transform was used for the first time to quantify the frequency, peak time, and power spectrum of the OP components of the adults’ ERG, providing more information than other wavelet transforms.

In [19], the aim was to classify glaucomatous and healthy sectors based on differences in frequency content within adults’ ERG using the Morlet wavelet transform and potentially the CWT. This approach could improve discrimination between normal and abnormal waveform signals in optic nerve diseases, which is essential for accurate diagnosis and treatment.

Finally, in [20], the Gaussian wavelet was chosen for its convenience in pediatric and adult ERG semi-automatic parameter extraction and better time domain properties. However, challenges remain in achieving simultaneous localization in both the frequency and time domains, indicating a need for further improvement in wavelet analysis techniques.

In summary, the selection of the mother wavelet plays a crucial role in ERG analysis, and various factors should be taken into consideration to ensure an accurate and comprehensive analysis of the data.

**Table 1 sensors-23-05813-t001:** Comparative table of used mother wavelets for CWT and studied signals (subjects).

Year	First Author and Reference	Mother Wavelet	Number of Signals (Subjects)
2005	Penkala [16]	Morlet Wavelet, Ricker Wavelet	120 (N/A)
2007	Penkala [15]		102 (N/A)
2010	Barraco [21]	Ricker Wavelet	24 (N/A)
2011	Barraco [22]		N/A (10)
2011	Barraco [23]		N/A (10)
2014	Gauvin [19]	Morse Wavelet	N/A (40)
2014	Dimopoulos [18]	Morlet Wavelet	N/A (63)
2015	Miguel-Jiménez [24]		N/A (47)
2020	Ahmadieh [17]		N/A (36)
2022	Zhdanov [20]	Gaussian Wavelet	425 (N/A)

## 3. Materials and Methods

### 3.1. Dataset Balancing

In this study signals from the IEEEDataPort repository were used, which is a publicly accessible ophthalmic electrophysiological signals database [25]. The dataset encompasses three types of pediatric signals: Maximum 2.0 ERG Response, Scotopic 2.0 ERG Response, and Photopic 2.0 ERG Response. Table 2 presents the Unbalanced Dataset column, which shows the number of signals in the dataset that belong to the healthy and unhealthy classes. The table reveals that the classes are imbalanced. To address this issue, the Imbalanced-learn package [26] was chosen as the solution, which has been utilized by researchers to solve such class imbalance issues. It is noteworthy that only pediatric signals were utilized, as they are the most representative and obviate the need for artificially generated signals.

An under-sampling technique was employed using the AllKNN function from the Imbalanced-learn package [26,27]. The AllKNN function uses the nearest neighbor algorithm to identify samples that contradict their neighborhood. The classical significant features of ERG signals were used as input to this function. To ensure the effectiveness of the nearest neighbor algorithm, the choice of the number of nearest neighbors to be considered is crucial. In our study, we use 13 as the number of nearest neighbors to achieve the desired class balance. The hyperparameter is chosen empirically: higher and lower values of nearest neighbors either remove too much data or do not remove it enough. The pairplot of the ERG signals distributions, presented in Figure 2, illustrates the results of this under-sampling technique, where orange and blue colors correspond to the healthy and unhealthy classes, respectively. It should be noted that the Scotopic signals were already balanced and did not require any under-sampling.

Thus, dataset balancing was implemented. Table 2 presents the distribution of healthy and unhealthy subjects within a balanced dataset. In this work, we use a balanced dataset for training experiments.

### 3.2. Training Pipeline

Figure 3 shows the training pipeline encompassing five distinct stages. During the Initial stage, the ERG signal dataset that is acquired in a time-domain representation is balanced. Subsequently, at the Transformation stage, the signal undergoes wavelet transformation, leading to a frequency-time representation, and is then stored in image classification dataset format.

Further, we split the train and the test subsets: undersampling the test set can lead to a biased evaluation of a model’s performance, which could be detrimental in real-world scenarios. Therefore, the test set represents the real-world distribution of healthy and unhealthy samples with an 85:15 ratio to the training subset.

At the following stages of Training and Cross-validation, the wavelet scalogram images are subjected to classification utilizing the training and validation datasets. Ultimately, the efficiency of the image classification process is assessed in the Evaluation stage using the balanced metrics.

#### 3.2.1. Data Preprocessing

The dataset under investigation contains signals comprising of 500 entries each, alongside their corresponding target (diagnosis). To perform an analysis, CWT was carried out on each signal using the PyWavelets library [28]. The base functions employed in this study were the commonly used ones, namely Ricker, Morlet, Gaussian Derivative, Complex Gaussian Derivative, and Shannon. The scaling parameters were adjusted to generate 512 × 512 gray-scale images.

#### 3.2.2. Baseline

The training was independently performed on five widely used architectures in the field of deep learning: VGG-11, ResNet-50, DensNet-121, ResNext-50 and Vision Transformer. The choice of models above was based on their popularity and proven effectiveness in the field of image classification. These models have been widely used in various computer vision tasks, and their performance has been thoroughly evaluated on standard datasets such as ImageNet [29,30,31]. In particular, using Vision Transformer, it is possible to investigate the effectiveness of this newer architecture compared to the more established models [32].

VGG-11 is one of the most popular pre-trained models for image classification. Introduced in the ILSVRC 2014 Conference, it remains the model to beat even today [33]. VGG-11 contains seven convolutional layers, each followed by a ReLU activation function, and five max pooling operations.

Residual Network (ResNet) is a specific type of convolutional neural network introduced in the paper “Deep Residual Learning for Image Recognition” by He Kaiming, et al., 2016 [34]. ResNet-50 is a 50-layer CNN that consists of 48 convolutional layers, one MaxPool layer, and one average pool layer.

ResNext-50 is a simple, highly modularized network architecture for image classification. It was constructed by repeating a building block that aggregates a set of transformations with the same topology [35].

DenseNet name refers to Densely Connected Convolutional Networks developed by Gao Huang, et al. in 2017 [36]. In this work, we used DensNet-121 that consists of 120 Convolutions and 4 AvgPool layers.

Vision Transformer is a model for image classification that employs a Transformer-like architecture over patches of the image. An image is split into patches with fixed size, each of the patches then linearly embedded, position embeddings are added, and the resulting sequence of vectors is fed to Transformer encoder [37]. In the current work, we used ViT_Small_r26_s32_224 model, pre-trained on a large collection of images in a supervised fashion, namely ImageNet-21k, at a resolution of 224 × 224 pixels.

We used ADAM optimization with 0.001 initial learning rate. Each model was trained until convergence using the early stopping criteria on the validation loss with batch size of 16 on a single NVIDIA V100.

#### 3.2.3. Loss Function

The loss function plays a critical role in deep learning. In this work, we utilize the most commonly used Cross-entropy loss function for classification tasks [38], which represents negative log-likelihood of a Bernoulli distribution (Equation 1):(1)$$CE\left(\tilde{y},\hat{y}\right)=-\frac{1}{N}\sum_{i=1}^{N}{\tilde{y}}_{i}\log({\hat{y}}_{i}),$$
where

$\tilde{y}$—one-hot encoded ground truth distribution,$\hat{y}$—predicted probability distribution,*N*—the size of the training set.

#### 3.2.4. Data Augmentation

The incorporation of data augmentation techniques during the training process leads to an augmentation of the distributional variability of input images. This augmentation is known to enhance the resilience of models by increasing their capacity to perform well on a wider range of inputs. Given the characteristics of our dataset, we opted to apply exclusively geometric transformations such as random cropping, vertical flipping, and image translation to the images under consideration.

#### 3.2.5. Cross Validation

Cross-validation is a resampling method that is employed to assess the effectiveness of deep learning models on a dataset with limited samples. The technique entails partitioning the dataset into k groups. To account for the limited nature of our dataset and facilitate a more objective evaluation of the trained models, we applied a five-fold cross-validation strategy in the present study. The test subset was first separated according to the real-world distribution of healthy and unhealthy clinical patients for each type of ERG response. The remaining shuffled training subset was then divided into five folds of which one is used for validation and four for training. The process is repeated for five experiments, using every fold once as the validation set.

#### 3.2.6. Evaluation

For each experiment, a confusion matrix was constructed using the test dataset, and the evaluation metrics were subsequently computed [39]. This approach enabled the accurate assessment of the model’s performance across different folds, thus ensuring a more comprehensive evaluation. Additionally, the confusion matrix provides a detailed overview of the model’s performance, highlighting the number of correct and incorrect predictions made by the model.

For a complete understanding of the model performance, several metrics were computed: Precision, Recall, and F1 score [40]:(2)$$Precision=\frac{TP}{TP+FP},$$
(3)$$Recall=\frac{TP}{TP+FN},$$
(4)$$F1=\frac{2*Precision*Recall}{Precision+Recall},$$
where

TP=TruePositive,FP=FalsePositive,FN=FalseNegative.

Since the test subset reflects the real-world distribution and is not balanced, we should consider Balanced Accuracy [41]:(5)$$Balanced\;Accuracy=\frac{Sensitivity+Specificity}{2},$$
where
(6)$$Sensitivity=Recall=\frac{TP}{TP+FN},$$
(7)$$Specificity=\frac{TN}{TN+FP}.$$

## 4. Results

Figure 4 shows box-plot distributions of classification accuracy where a is Maximum 2.0 ERG Response, b—Scotopic 2.0 ERG Response, c—Photopic 2.0 ERG Response.

The findings from Figure 4a suggest that the Ricker Wavelet combined with the Vision Transformer produces the highest classification accuracy for ERG wavelet scalograms. Specifically, the median balanced accuracy value is 0.83, with the upper quartile being 0.85 and the lower quartile being 0.8. In contrast, when utilizing the Shannon Wavelet, the median accuracy value is 0.8, with the upper quartile at 0.82 and the lower quartile at 0.77. Additionally, for the Morlet Wavelet, the median balanced accuracy value is 0.78, with the upper quartile at 0.81 and the lower quartile at 0.77.

The findings from Figure 4b suggest that the Ricker Wavelet combined with the Vision Transformer produces the highest classification accuracy for ERG wavelet scalograms. Specifically, the median balanced accuracy value is 0.85, with the upper quartile being 0.87 and the lower quartile being 0.84. In contrast, when utilizing the Morlet Wavelet, the median accuracy value is 0.82, with the upper quartile at 0.86 and the lower quartile at 0.76. Additionally, for the Gaussian Wavelet, the median balanced accuracy value is 0.78, with the upper quartile at 0.79 and the lower quartile at 0.77.

The findings from Figure 4c suggest that the Ricker Wavelet combined with the Vision Transformer produces the highest classification accuracy for ERG wavelet scalograms. Specifically, the median balanced accuracy value is 0.88, with the upper quartile being 0.92 and the lower quartile being 0.88. In contrast, when utilizing the Shannon Wavelet, the median accuracy value is 0.86, with the upper quartile at 0.87 and the lower quartile at 0.83. Additionally, for the Morlet Wavelet, the median balanced accuracy value is 0.85, with the upper quartile at 0.93 and the lower quartile at 0.79.

More detailed values of the metrics are given in the Appendix A of the article in Table A1, Table A2 and Table A3.

The results from Figure 4 provide valuable insights into the accuracy of ERG wavelet scalogram classification. In all three cases, the Ricker Wavelet combined with the Vision Transformer yielded the highest median accuracy values, demonstrating the effectiveness of this combination. Additionally, the upper and lower quartile values further support the superiority of this approach, showing consistently high accuracy levels. However, it is important to note that the performance of other wavelet types should not be overlooked. For example, when utilizing the Shannon Wavelet in Figure 4b, a median balanced accuracy value of 0.82 was achieved, which is still a relatively high level of accuracy. Similarly, in Figure 4c, the Morlet Wavelet produced a median balanced accuracy value of 0.85, which is also noteworthy. Overall, these findings suggest that the selection of the mother wavelet plays a critical role in determining the accuracy of the classification of ERG wavelet scalograms. By carefully selecting the most appropriate wavelet type and transformer architecture, it may be possible to achieve even higher accuracy levels, thereby advancing our understanding of ERG wavelet scalogram classification.

The selection of the mother wavelet in ERG analysis is of utmost importance as it directly influences the quality and interpretability of the results. Choosing an appropriate mother wavelet requires careful consideration of various factors to ensure accurate and comprehensive data analysis. Despite the abundance of literature on ERG analysis, there is a lack of a clearly formulated motivation for selecting a specific mother wavelet. Existing sources often fail to provide explicit reasoning or guidelines for choosing one wavelet over another in the context of ERG analysis. This gap in the literature hinders researchers from making informed decisions regarding the selection of the most suitable mother wavelet for their ERG data analysis, highlighting the need for further research and guidance in this area.

The dataset comprised signals with 500 entries each, and CWT was applied using the PyWavelets library to generate 512 × 512 gray-scale images. The CWT was performed using different base functions, including Ricker, Morlet, Gaussian Derivative, Complex Gaussian Derivative, and Shannon.

Deep learning architectures, namely VGG-11, ResNet-50, DenseNet-121, ResNext-50, and Vision Transformer, were independently trained on the dataset to establish baselines for performance comparison. These models have been extensively evaluated in computer vision tasks and have shown effectiveness in image classification. Training was carried out until convergence using the ADAM optimization with an initial learning rate of 0.001 and a batch size of 16 on a single NVIDIA V100.

The performance of the models was evaluated using cross-validation with a five-fold strategy to account for the limited sample size. The dataset was divided into training and test subsets, and the training subset was further partitioned into five folds for validation. Confusion matrices were constructed using the test dataset, enabling the computation of evaluation metrics such as precision, recall, F1 score, and balanced accuracy. This approach provided a comprehensive assessment of the models’ performance across different folds and allowed for a detailed analysis of correct and incorrect predictions made by the models.

The results demonstrated the effectiveness of the deep learning models in classifying the signals and diagnosing the corresponding conditions. Overall, the Vision Transformer model exhibited the highest performance, achieving the highest precision, recall, F1 score, and balanced accuracy among the tested architectures. The VGG-11, ResNet-50, and DenseNet-121 models also displayed strong performance, while the ResNext-50 model achieved slightly lower metrics. These findings highlight the potential of deep learning models, particularly the Vision Transformer architecture, in analyzing signal datasets and facilitating accurate diagnoses. The study’s results contribute to the understanding of the applicability of deep learning techniques in medical diagnostics and pave the way for future research in this domain.

## 5. Discussion

The choice of wavelet for ERG signal analysis depends on the waveform characteristics, with different wavelets having varying frequencies and temporal resolutions. An optimal wavelet should possess effective noise suppression capabilities [42], accurately capture transient and sustained components of the ERG signal, and provide interpretable coefficients for feature identification. Computational efficiency is important for handling large datasets and real-time applications. Additionally, the familiarity and expertise of the researcher or clinician in interpreting specific wavelets can enhance the accuracy and efficiency of the analysis. Careful wavelet selection is crucial to ensure reliable and meaningful results in clinical and research settings [43].

The Ricker Wavelet yielded the highest median accuracy values for ERG wavelet scalogram classification due to the following potential reasons:Wavelet characteristics: the specific properties of the Ricker Wavelet, including its shape and frequency properties, align well with the features present in ERG wavelet scalograms, leading to improved accuracy in classification compared to other wavelet types.Noise suppression capabilities: the Ricker Wavelet demonstrates superior noise suppression capabilities, effectively reducing unwanted noise in ERG wavelet scalograms while preserving important signal components, resulting in enhanced accuracy.Time-frequency localization: the Ricker Wavelet excels in accurately localizing transient and sustained components of ERG waveforms across different time intervals, enabling better capture and representation of crucial temporal features, thereby increasing the discriminative power of the wavelet in classifying ERG responses.

It should be noted that the present study utilized a limited set of signals to identify the most appropriate mother wavelet for ERG analysis. Nevertheless, the sample was well-balanced, which lends confidence to the relatively stable classification outcomes.

In comparison to electrophysiological data, which typically contain numerous parameters describing motor function, ERG analysis necessitates the addition of significant parameters to ensure the efficient classification of specific states. As ERG analysis involves only four parameters [44], insufficient for precise diagnosis, expanding the feature space via continuous wavelet transform in the frequency-time domain is essential.

Selected neural networks may exhibit superior performance when trained on larger datasets. For instance, the accuracy distribution of the Transformer model displays a wide range [45]; however, this variability would likely be reduced with an increase in the size of the training dataset. Moreover, it was essential to divide and keep the test data without modification based on the distribution observed in real-world scenarios, which affected the quantity of training data that is available.

As the ERG signals are equipment- and intensity-specific, it is important to exercise caution when combining different datasets. An area of research that holds promise involves the creation of synthetic signals, which can augment the available training data.

This research investigates the potential of AI algorithms in accurately classifying eye diseases and acknowledges their role as supportive tools to medical specialists. While the algorithms demonstrate good accuracy, we emphasize the indispensability of specialist involvement. The complex nature of human health, the significance of empathetic care, and the unparalleled decision-making capabilities of doctors underscore their ongoing essential role in delivering comprehensive and holistic medical care. Classification algorithms can serve as clinical decision support systems, enhancing physicians’ expertise and facilitating more efficient and effective healthcare delivery [46,47].

### Limitations

Equipment limitations: The utilization of only the Tomey EP-1000 equipment for ERG registration introduces a potential limitation to the generalizability of the model’s results. As different equipment may exhibit variations in signal acquisition and measurement precision, the model’s performance and outcomes may differ when applied with alternative ERG registration devices. Furthermore, the use of a corneal electrode during ERG registration does not entirely eliminate the possibility of electrooculogram-induced noise stemming from the involuntary movement of the eye muscles [3]. Consequently, the presence of such noise may impact the accuracy and reliability of the obtained ERG signals, influencing the model’s performance.

Study protocol considerations: The employment of specific ERG protocols, namely Maximum 2.0 ERG Response, Scotopic 2.0 ERG Response, and Photopic 2.0 ERG Response, within this investigation, ensures the acquisition of ERG recordings with optimal quality on the employed equipment [48]. However, it is crucial to acknowledge that altering the study protocol, such as modifying the brightness or timing of light stimuli, could yield varying results when utilizing the model. Changes in these protocol parameters may introduce variations in the recorded ERG signals, potentially impacting the model’s predictive performance and its ability to generalize to different experimental conditions or protocols.

Dataset limitations: The dataset used in this study comprises data from both healthy subjects and subjects with retinal dystrophy [49]. While the inclusion of healthy subjects provides a baseline for comparison, the focus of the model’s training and evaluation is on the detection and diagnosis of retinal dystrophy. Therefore, it is important to note that this specific model is designed and optimized for the diagnosis of retinal dystrophy and may not be applicable for the diagnosis of other diseases or conditions. The model’s performance and generalizability to other diseases should be assessed separately using appropriate datasets and evaluation protocols.

Neural network feature limitations: While neural networks have shown improved performance with larger datasets, it is essential to consider the limitations associated with the available data. The accuracy of the chosen neural networks, especially the Transformer, increases with an increase in the training subset. However, the available datasets are limited, and the different nature of the origin of the signals makes it difficult to combine such datasets.

To overcome equipment limitations in ERG recordings, the use of Erg-Jet electrodes can be considered [50]. These electrodes help minimize noise caused by the movement of eye muscles, thus improving the quality of the recorded signals. Additionally, standardizing the study protocol across different equipment setups can help address the issue of using different equipment [44].

Regarding study protocol limitations, the adoption of standardized protocols ensures a uniform approach to ERG recordings. Furthermore, presenting detailed statistical data about the dataset used in the study helps address study protocol limitations by providing transparency and enabling researchers to evaluate the robustness and generalizability of the findings.

To mitigate dataset limitations, it is crucial to expand the ERG dataset and include a broader range of retinal diseases. By increasing the size and diversity of the dataset, researchers can enhance the representativeness of the data and improve the model’s ability to generalize to various clinical scenarios. Incorporating additional retinal diseases beyond the scope of the current study would enable a more comprehensive understanding of the model’s performance and its applicability to a wider range of clinical conditions. Expanding the dataset can also help identify potential subgroups or rare conditions that may have specific ERG characteristics, contributing to the advancement of knowledge in the field. A promising development direction is the generation of synthetic ERG signals: combining the mathematical model of the signal and Generative Adversarial Networks [51]. This will increase the training subset with signals of a similar origin.

## 6. Conclusions

The results of this study indicate that the combination of Ricker Wavelet combined with Vision Transformer consistently achieves the highest median balanced accuracy values across all three ERG responses: 0.83, 0.85, and 0.88 consequently. The robust upper and lower quartile values provide compelling evidence for the superiority of this combination, consistently demonstrating high accuracy levels. However, it is important to acknowledge that other wavelet types also yield relatively high accuracy levels and should not be disregarded. These findings underscore the critical role of selecting an appropriate mother wavelet in determining the accuracy of ERG wavelet scalogram classification. Careful consideration of the wavelet type and transformer architecture holds significant potential for attaining even higher levels of accuracy. Overall, this study offers valuable insights into the effectiveness of different wavelet-model combinations, thereby contributing to the precise classification of pediatric ERG signals and advancing the field of healthcare.

The findings of this study will be utilized to develop an AI-based decision support system in ophthalmology, leveraging the insights gained from ERG wavelet scalogram classification. This system aims to enhance ophthalmology-related applications by incorporating accurate and efficient analysis of ERGs. Furthermore, the results of this study may hold value for manufacturers of electrophysiological stations used in ERGs. The understanding of which wavelet types, such as the Ricker Wavelet, yield superior classification accuracy may guide the development and optimization of electrophysiological stations, enabling them to provide more reliable and advanced diagnostic capabilities in the field of ophthalmology.

## Figures and Tables

**Figure 1 sensors-23-05813-f001:**
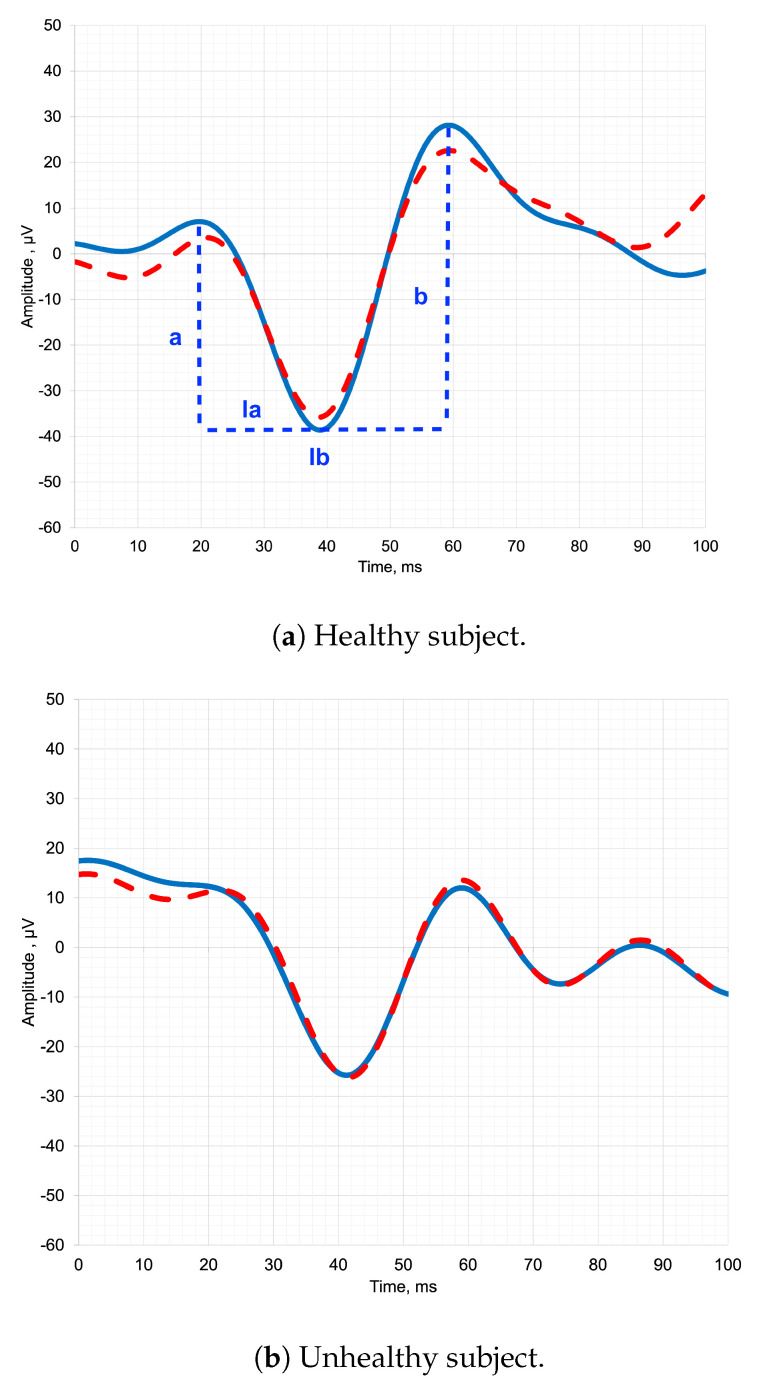
Dark-adapted full-field electroretinogram: (**a**) healthy subject; (**b**) unhealthy subject. Blue and red lines show two different examples of the ERG signal.

**Figure 2 sensors-23-05813-f002:**
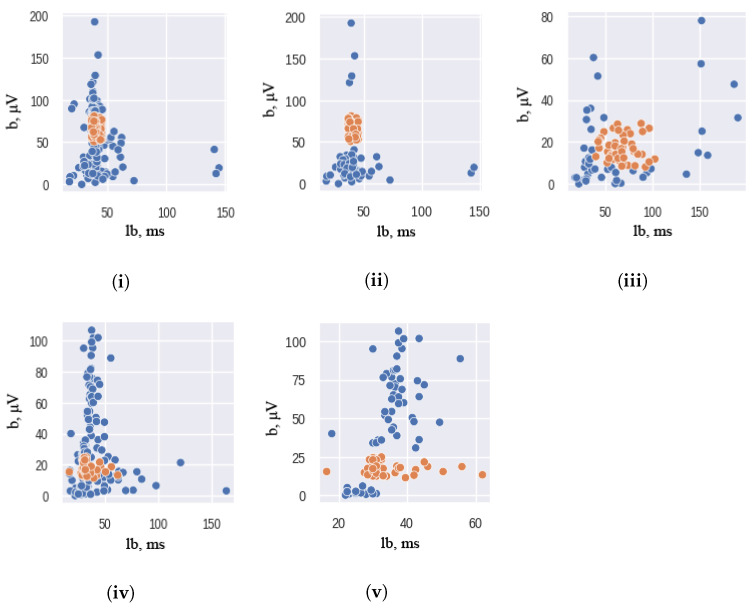
Scatterplot Visualizations of the ERG signal classical features: Maximum 2.0 ERG Response—before (**i**) and after (**ii**) under—sampling; Scotopic 2.0 ERG Response—(**iii**); Photopic 2.0 ERG Response—before (**iv**) and after (**v**) under-sampling. Here b—is the amplitude of the b-wave (µV), lb is the latency of the b-wave (ms). Orange and blue colors correspond to the healthy and unhealthy classes respectively.

**Figure 3 sensors-23-05813-f003:**
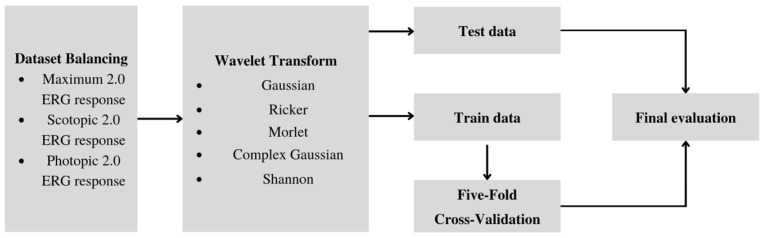
Training Pipeline: dataset balancing, wavelet transformation, splitting the data on test and train datasets, Cross-Validation and final evaluation of the model.

**Figure 4 sensors-23-05813-f004:**
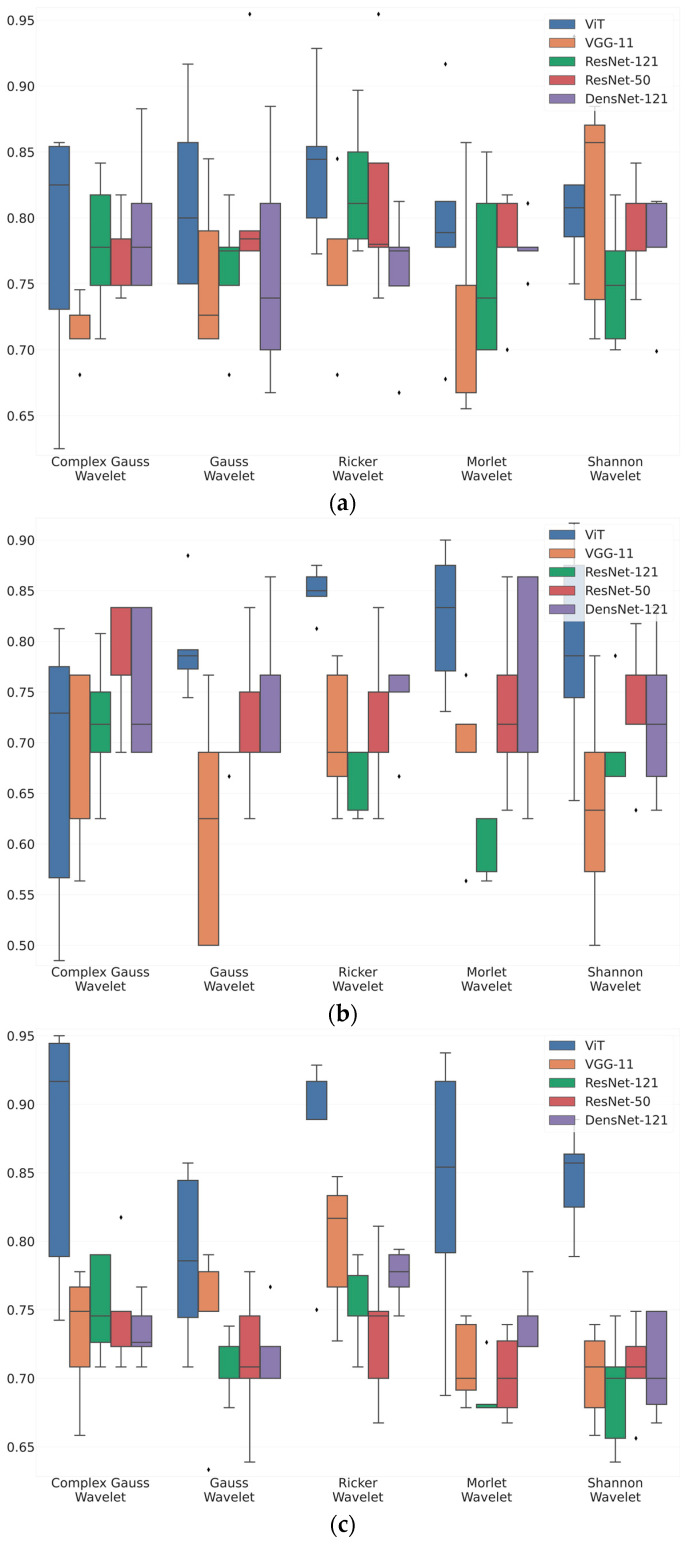
Box-plot distributions of classification accuracy: (**a**) Maximum 2.0 ERG Response; (**b**) Scotopic 2.0 ERG Response; (**c**) Photopic 2.0 ERG Response.

**Table 2 sensors-23-05813-t002:** Comparative table of ERG signals before and after balancing.

Unbalanced Dataset		Balanced Dataset	
**Healthy**	**Unhealthy**	**Healthy**	**Unhealthy**
Maximum 2.0 ERG Response			
60	143	60	62
Scotopic 2.0 ERG Response			
48	52	48	52
Photopic 2.0 ERG Response			
68	171	68	63

## Data Availability

Zhdanov, A.E.; Dolganov, A.Y.; Borisov, V.I.; Lucian, E.; Bao, X.; Kazaijkin, V.N.; Ponomarev, V.O.; Lizunov, A.V.; Ivliev, S.A. 355 OculusGraphy: Pediatric and Adults Electroretinograms Database, 2020. https://dx.doi.org/10.21227/y0fh-5v04. (accessed on 29 November 2022).

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
