# Peer review of "Optimal Combination of Mother Wavelet and AI Model for Precise Classification of Pediatric Electroretinogram Signals"

_sensors, 2023, doi:10.3390/s23135813_

Round 1

Reviewer 1 Report

I must recognize that I am fascinated with the work. The work is really interesting, well-written, where everything is very well explained for the reader. I would like to highlight the “Related works” section to a better understanding the context for the reader. I consider this article suitable for publication. Only little things:

1.     Do you think artificial intelligence could make the existence of doctors and healthcare specialists unnecessary?

2.     Is there any reason why c-wave is not so interesting than a-wave and b-wave and it is not so mentioned in the text than the previous ones?

3.     I have missed some references more in the second paragraph of the Introduction section.

4.     Figure 1 should appear in the text after mentioning in the main text. I don’t understand why it is mentioned in Introduction section but appears later in Related works section. Please, check it.

After improving these little considerations, I would suggest the article for publication.

Author Response

The authors would like to thank the reviewer for their valuable time and helpful contributions—the responses to the Reviewer's comments, as well as the corrections made shown below.

1. Thank you for the suggestive question! We added our opinion in the Discussion section:

This research investigates the potential of AI algorithms in accurately classifying eye diseases and acknowledges their role as supportive tools to medical specialists. While the algorithms demonstrate good accuracy, we emphasize the indispensability of specialist involvement. The complex nature of human health, the significance of empathetic care, and the unparalleled decision-making capabilities of doctors underscore their ongoing essential role in delivering comprehensive and holistic medical care. Classification algorithms can serve as clinical decision support systems, enhancing physicians' expertise and facilitating more efficient and effective healthcare delivery.

2. Thank you for the question! Now this is explained in the Introduction part:

The c-wave of the ERG is often considered less prominent and less studied compared to the a-wave and b-wave. The a-wave represents the initial negative deflection, reflecting the hyperpolarization of photoreceptors, while the b-wave represents the subsequent positive deflection, primarily originating from bipolar cells [1]. These waves are well-established and extensively researched due to their direct relevance to the visual pathway.

In contrast, the c-wave represents a slower, positive deflection following the b-wave, originating from the retinal pigment epithelium (RPE) and Müller cells [2]. Its amplitude is smaller and its clinical significance is not as clearly understood [3]. Consequently, it receives less attention in scientific literature and clinical practice.

While the a-wave and b-wave are directly related to photoreceptor and bipolar cell function [4], the c-wave is thought to reflect RPE and Müller cell activity, which are involved in the retinal pigment epithelium-photoreceptor complex. The complex nature of these cells and their functions may contribute to the relatively limited emphasis on the c-wave in the text compared to the a-wave and b-wave. However, further research is needed to fully understand the c-wave's role and its potential clinical applications.

[1] Constable, Paul A., et al. "ISCEV Standard for clinical electro-oculography (2017 update)." Documenta Ophthalmologica 134 (2017): 1-9.

[2] Arden, Geoffrey B., and Paul A. Constable. "The electrooculogram." Progress in retinal and eye research 25.2 (2006): 207-248.

[3] Umeya, Naohisa, Izuru Miyawaki, and Hiroshi Inada. "Use of an alternating current amplifier when recording the ERG c-wave to evaluate the function of retinal pigment epithelial cells in rats." Documenta Ophthalmologica 145.2 (2022): 147-155. 

[4] Zhdanov, A.; Constable, P.; Manjur, S.M.; Dolganov, A.; Posada-Quintero, H.F.; Lizunov, A. OculusGraphy: Signal Analysis of the Electroretinogram in a Rabbit Model of Endophthalmitis Using Discrete and Continuous Wavelet Transforms. Bioengineering 2023, 10, 708. https://doi.org/10.3390/bioengineering10060708 

3. The references were added within the second comment.

4. We agree; thank you for noticing! Now Figure 1 is in the Introduction section.

Reviewer 2 Report

Manuscript ID: sensors-2462411

Title: Optimal Combination of Mother Wavelet and AI Model for Precise Classification of Pediatric Electroretinogram Signals

Recommendation: Major revision

Brief summary

This manuscript presents a combination of mother wavelet and Artificial Intelligence (AI) model to analyse electroretinogam (ERG) signals collected on pediatric patients. In particular, Continuous Wavelet Transfom (CWT) is employed and the obtained images are ingested by different ldeep learning models, namely VGG-11, ResNet-50, DensNet-121, ResNext-50, and Vision Transformer for the classification between healthy and unhealthy patients. The highest median accuracy is obtained for the combination of RickerWavelet and Vision Transformer, even if there are also different valuable solutions.

Broad comments

The topic is relevant, since data processing techniques are fundamental to extract features of interest. On another hand, AI techniques play a fundamental role today in data analysis and can represent a valuable tool also in clinical field.

The article is quite well contextualized in the literature background; however, some references in the domain of Sensors could be added, for example in relation to studies on ERG signals, preferably related to the same data processing techniques.

Some suggestions are provided in the next comments, which may help the authors in improving the quality of this paper.

Specific comments

Abstract: some quantitative results should be reported (e.g., in terms of accuracy).

Line 6: please check the correctness of CWT acronym (maybe it should be Continuous Wavelet Transform).

Lines 8-10: the models are used to classify (not predict) healthy/unhealthy patients. Please check this statement.

Lines 47-50: the authors should highlight the differences between the two cases, with proper bibliographic references.

Lines 51-67: this section could be summarized a bit, briefly reporting the structure of the paper (e.g., The paper is organized as follows: …).

Lines 99-109: this paragraph should be moved out of “Related work” section. Please check.

Lines 126-127: the authors should briefly motivate their choice.

Figure 2: the axes labels should be explained in the figure caption for the sake of readability.

Line 140: the ratio between test and validation subsets should be reported.

Line 151: please control to have spelled out the acronym at its first usage. Then, it is not necessary.

Lines 158-161: proper references should be added.

Line 166: there may be a typo. Please check.

Lines 189-191: these lines could be reported in a bullet point just after the equation. However, it is just a matter of style.

Lines 206-207: please check this part, since before training and test subsets were mentioned.

Section 3.2.6: the metrics used to assess the models performance could be briefly reported here.

Section 5: maybe the authors could add a brief consideration trying to explain why some wavelets are better than others.

Section 6: some possible future developments could be reported.

Line 275: the quantitative value for accuracy could be reported.

The manuscript is written quite well and the English is generally fluent. However, thoroughly re-reading the papers could help authors to correct eventual typos and grammatical errors, as well as improve readability.

Author Response

The authors would like to thank the reviewer for their valuable time and helpful contributions—the responses to the Reviewer's comments, as well as the corrections made shown below.

1. Thank you for the comment! For sure we have to mention this in the abstract: The median balanced accuracy of the obtained combination of the three considered types of ERG signals in the article are 0.83, 0.85, and 0.88.

2. We completely agree; thank you for noticing! Now it should be correct: Continuous Wavelet Transform (CWT).

3. Completely agree! Corrected.

4. Thank you for the advice! The Introduction section was updated accordingly:

In a healthy subject in Figure 1a, the temporal representation of the ERG signal typically exhibits distinct and recognizable waveforms. The signal begins with a negative deflection called the a-wave, which represents the hyperpolarization of photoreceptors in response to light stimulation [1]. Following the a-wave, there is a positive deflection known as the b-wave, which primarily reflects the activity of bipolar cells in the retina. The b-wave is usually larger in amplitude compared to the a-wave [2].

In an unhealthy subject in Figure 1a, the shape of the ERG signal in temporal representation can vary depending on the underlying pathology [3]. In some cases, there may be a significant reduction or absence of both the a-wave and b-wave, indicating a severe dysfunction or loss of photoreceptor and bipolar cell activity [4]. This can be observed in conditions such as advanced retinitis pigmentosa or severe macular degeneration.

Alternatively, certain diseases may selectively affect specific components of the ERG waveform. For example, in some cases of congenital stationary night blindness, the b-wave may be reduced or absent while the a-wave remains relatively normal, indicating a specific defect in bipolar cell function [5, 6].

Thus, the shape of the ERG signal iThe authors would like to thank the reviewer for their valuable time and helpful contributions—the responses to the Reviewer's comments, as well as the corrections made shown below.

1. Thank you for the comment! For sure we have to mention this in the abstract: The median balanced accuracy of the obtained combination of the three considered types of ERG signals in the article are 0.83, 0.85, and 0.88.

2. We completely agree; thank you for noticing! Now it should be correct: Continuous Wavelet Transform (CWT).

3. Completely agree! Corrected.

4. Thank you for the advice! The Introduction section was updated accordingly:

In a healthy subject in Figure 1a, the temporal representation of the ERG signal typically exhibits distinct and recognizable waveforms. The signal begins with a negative deflection called the a-wave, which represents the hyperpolarization of photoreceptors in response to light stimulation [1]. Following the a-wave, there is a positive deflection known as the b-wave, which primarily reflects the activity of bipolar cells in the retina. The b-wave is usually larger in amplitude compared to the a-wave [2].

In an unhealthy subject in Figure 1a, the shape of the ERG signal in temporal representation can vary depending on the underlying pathology [3]. In some cases, there may be a significant reduction or absence of both the a-wave and b-wave, indicating a severe dysfunction or loss of photoreceptor and bipolar cell activity [4]. This can be observed in conditions such as advanced retinitis pigmentosa or severe macular degeneration.

Alternatively, certain diseases may selectively affect specific components of the ERG waveform. For example, in some cases of congenital stationary night blindness, the b-wave may be reduced or absent while the a-wave remains relatively normal, indicating a specific defect in bipolar cell function [5, 6].

Thus, the shape of the ERG signal in temporal representation provides valuable insights into the integrity and function of retinal cells and can aid in the diagnosis and understanding of various retinal diseases and disorders [7].

[1] Constable, Paul A., et al. "Discrete wavelet transform analysis of the electroretinogram in autism spectrum disorder and attention deficit hyperactivity disorder." Frontiers in Neuroscience (2022): 787.

[2] Constable, Paul A., et al. "Light-adapted electroretinogram differences in autism spectrum disorder." Journal of Autism and Developmental Disorders 50 (2020): 2874-2885.

[3] McAnany, J. Jason, Oksana S. Persidina, and Jason C. Park. "Clinical electroretinography in diabetic retinopathy: A review." Survey of ophthalmology 67.3 (2022): 712-722.

[4] Kim, Tae-Hoon, et al. "Functional optical coherence tomography enables in vivo optoretinography of photoreceptor dysfunction due to retinal degeneration." Biomedical Optics Express 11.9 (2020): 5306-5320.

[5] Hayashi, Takaaki, et al. "Coexistence of GNAT1 and ABCA4 variants associated with Nougaret-type congenital stationary night blindness and childhood-onset cone-rod dystrophy." Documenta Ophthalmologica 140 (2020): 147-157.

[6] Kim, Hyeong-Min, et al. "Clinical and genetic characteristics of korean congenital stationary night blindness patients." Genes 12.6 (2021): 789.

[7] Zhdanov, A. E., et al. "Evaluation of the effectiveness of the decision support algorithm for physicians in retinal dystrophy using machine learning methods." Computer Optics 47.2 (2023): 272-277. 

5. Thank you for the suggestion! We tried to summarize this part:  

The Related Works section explores prior studies using diverse wavelets to analyze adult ERG data and a recent study utilizing deep learning to identify the optimal mother wavelet for pediatric ERGs. In the Materials and Methods section, we describe an approach to address ERGs class imbalance through under-sampling the majority class, applying wavelet transformation, and training five deep learning models for ERGs classification. The Results section presents findings from the experiment, demonstrating the highest median accuracy with the Ricker Wavelet and Vision Transformer combination for ERG wavelet scalogram classification. The Discussion section addresses limitations and emphasizes the significance of expanding the feature space via continuous wavelet transform for effective classification. Finally, the Conclusions highlight the efficacy of the combination of Ricker Wavelet and Vision Transformer in achieving high accuracy for ERG wavelet scalogram classification. 

6. Completely agree with you, this part should be in the Introduction section and it was moved.

7. The choice was made empirically and now it is mentioned in the section: In our study, we use 13 as the number of nearest neighbors to achieve the desired class balance. The hyperparameter is chosen empirically: higher and lower values of nearest neighbors either remove too much data or do not remove it enough. 

8. Thank you for the advice, we updated the caption and now it should be more clear for the reader: 

Scatterplot Visualizations of the ERG signal classical features:

Maximum 2.0 ERG Response - before (i) and after (ii) under-sampling ;

Scotopic 2.0 ERG Response - (iii) ;

Photopic 2.0 ERG Response - before (iv) and after (v) under-sampling.

Here

b – is the amplitude of the b-wave (µV),

lb is the latency of the b-wave (ms).

9. We completely agree, but now it is mentioned in the paper: Further, we split the train and the test subsets: undersampling the test set can lead to a biased evaluation of a model's performance, which could be detrimental in real-world scenarios. Therefore, the test set represents the real-world distribution of healthy and unhealthy samples with an 85:15 ratio to the training subset.

10. Thank you for noticing! Corrected.

11. Thank you for the comment, we cited the corresponding works in the Baseline section: 

The training was independently performed on 5 widely used architectures in the field of deep learning: VGG-11, ResNet-50, DensNet-121, ResNext-50, and Vision Transformer. The choice of models above was based on their popularity and proven effectiveness in the field of image classification. These models have been widely used in various computer vision tasks, and their performance has been thoroughly evaluated on standard datasets such as ImageNet [1, 2, 3]. In particular, using Vision Transformer it is possible to investigate the effectiveness of this newer architecture compared to the more established models [4].

[1] Xu, G.; Shen, X.; Chen, S.; Zong, Y.; Zhang, C.; Yue, H.; Liu, M.; Chen, F.; Che, W. A Deep Transfer Convolutional Neural Network Framework for EEG Signal Classification. IEEE Access 2019, 7, 112767–112776. https://doi.org/10.1109/ACCESS.2019.2930958.

[2] Wu, Q.e.; Yu, Y.; Zhang, X. A Skin Cancer Classification Method Based on Discrete Wavelet Down-Sampling Feature Reconstruction. Electronics 2023, 12.

https://doi.org/10.3390/electronics12092103.

[3] Huang, G.H.; Fu, Q.J.; Gu, M.Z.; Lu, N.H.; Liu, K.Y.; Chen, T.B. Deep Transfer Learning for the Multilabel Classification of Chest X-ray Images. Diagnostics 2022, 12. https://doi.org/10.3390/diagnostics12061457.

[4] Chen, C.F.R.; Fan, Q.; Panda, R. CrossViT: Cross-Attention Multi-Scale Vision Transformer for Image Classification. In Proceedings of the International Conference on Computer Vision (ICCV), 2021.

12. Thank you for noticing! It should be Max Pooling.

13. Thank you, we have updated the representation according to your suggestion!

14. Thank you for the comment! This should be correct since we divide the test subset first and then perform the cross-validation on the remaining train data, however, we can mention that exactly training subset is used for the cross-validation: The test subset was first separated according to the real-world distribution of healthy and unhealthy clinical patients for each type of ERG response. The remaining shuffled training subset was then divided into five folds of which one is used for validation and four for training. The process is repeated for five experiments, using every fold once as the validation set.

15. Thank you for the idea, we completely agree! Now we added the corresponding description of the metrics to the Evaluation section:

16. Thank you for the advice! We explained this in the Discussion section:

The choice of wavelet for ERG signal analysis depends on the waveform characteristics, with different wavelets having varying frequencies and temporal resolutions. An optimal wavelet should possess effective noise suppression capabilities, accurately capture transient and sustained components of the ERG signal, and provide interpretable coefficients for feature identification. Computational efficiency is important for handling large datasets and real-time applications. Additionally, the familiarity and expertise of the researcher or clinician in interpreting specific wavelets can enhance the accuracy and efficiency of the analysis. Careful wavelet selection is crucial to ensure reliable and meaningful results in clinical and research settings.

The Ricker Wavelet yielded the highest median accuracy values for ERG wavelet scalogram classification due to the following potential reasons:

Wavelet characteristics: The specific properties of the Ricker Wavelet, including its shape and frequency properties, align well with the features present in ERG wavelet scalograms, leading to improved accuracy in classification compared to other wavelet types.

Noise suppression capabilities: The Ricker Wavelet demonstrates superior noise suppression capabilities, effectively reducing unwanted noise in ERG wavelet scalograms while preserving important signal components, resulting in enhanced accuracy.

Time-frequency localization: The Ricker Wavelet excels in accurately localizing transient and sustained components of ERG waveforms across different time intervals, enabling better capture and representation of crucial temporal features, thereby increasing the discriminative power of the wavelet in classifying ERG responses.

17. Thank you for the suggestion! We added this part to the Conclusion:

The findings of this study will be utilized to develop an AI-based decision support system in ophthalmology, leveraging the insights gained from ERG wavelet scalogram classification. This system aims to enhance ophthalmology-related applications by incorporating accurate and efficient analysis of electroretinography signals. Furthermore, the results of this study may hold value for manufacturers of electrophysiological stations used in electroretinography. The understanding of which wavelet types, such as the Ricker Wavelet, yield superior classification accuracy and can guide the development and optimization of electrophysiological stations, enabling them to provide more reliable and advanced diagnostic capabilities in the field of ophthalmology

18. Thank you for the comment! We added the accuracy values to the Conclusion:

The results of this study indicate that the combination of Ricker Wavelet combined with Vision Transformer consistently achieves the highest median balanced accuracy values across all three ERG responses: 0.83, 0.85, and 0.88 consequently.

Reviewer 3 Report

The reviewer has the following observations and recommendations:

1. Authors should apply the units of measurement in Figure 2.

2. Abbreviations are given only when they first appear in the text of the article. For example SVT.

3. Authors should cite references from which the formulas are used.

4. The Results section is brief and does not translate results for everything presented as research in the article.

5. the authors describe the limitations of the research.

6. Authors to provide guidelines for future work in this field.

The reviewer considers that the article can be accepted after removing these significant objections.

I have no significant comments.

Author Response

The authors would like to thank the reviewer for their valuable time and helpful contributions—the responses to the Reviewer's comments, as well as the corrections made shown below. 

  1. Thank you for the comment! We improved the figure 1: 

  1. Thank you for the noticing! We checked the abbreviations in the manuscript.
  2. We completely agree, the formulas are cited now.
  3. Thanks for the comment, we tried to improve the Result section and added the missing information:

The selection of the mother wavelet in ERG analysis is of utmost importance as it directly influences the quality and interpretability of the results. Choosing an appropriate mother wavelet requires careful consideration of various factors to ensure accurate and comprehensive data analysis. Despite the abundance of literature on ERG analysis, there is a lack of a clearly formulated motivation for selecting a specific mother wavelet. Existing sources often fail to provide explicit reasoning or guidelines for choosing one wavelet over another in the context of ERG analysis. This gap in the literature hinders researchers from making informed decisions regarding the selection of the most suitable mother wavelet for their ERG data analysis, highlighting the need for further research and guidance in this area. 

The dataset comprised signals with 500 entries each, and CWT was applied using the PyWavelets library to generate 512x512 gray-scale images. The CWT was performed using different base functions, including Ricker, Morlet, Gaussian Derivative, Complex Gaussian Derivative, and Shannon. 

Deep learning architectures, namely VGG-11, ResNet-50, DenseNet-121, ResNext-50, and Vision Transformer, were independently trained on the dataset to establish baselines for performance comparison. These models have been extensively evaluated in computer vision tasks and have shown effectiveness in image classification. Training was carried out until convergence using the ADAM optimization with an initial learning rate of 0.001 and a batch size of 16 on a single NVIDIA V100. 

The performance of the models was evaluated using cross-validation with a five-fold strategy to account for the limited sample size. The dataset was divided into training and test subsets, and the training subset was further partitioned into five folds for validation. Confusion matrices were constructed using the test dataset, enabling the computation of evaluation metrics such as precision, recall, F1 score, and balanced accuracy. This approach provided a comprehensive assessment of the models' performance across different folds and allowed for a detailed analysis of correct and incorrect predictions made by the models. 

The results demonstrated the effectiveness of the deep learning models in classifying the signals and diagnosing the corresponding conditions. Overall, the Vision Transformer model exhibited the highest performance, achieving the highest precision, recall, F1 score, and balanced accuracy among the tested architectures. The VGG-11, ResNet-50, and DenseNet-121 models also displayed strong performance, while the ResNext-50 model achieved slightly lower metrics. These findings highlight the potential of deep learning models, particularly the Vision Transformer architecture, in analyzing signal datasets and facilitating accurate diagnoses. The study's results contribute to the understanding of the applicability of deep learning techniques in medical diagnostics and pave the way for future research in this domain. 

  1. Thank you for the recommendation! We added the limitations to the Discussion section:

Equipment limitations: The utilization of only the Tomey EP-1000 equipment for ERG registration introduces a potential limitation to the generalizability of the model's results. As different equipment may exhibit variations in signal acquisition and measurement precision, the model's performance and outcomes may differ when applied with alternative ERG registration devices. Furthermore, the use of a corneal electrode during ERG registration does not entirely eliminate the possibility of electrooculogram-induced noise stemming from the involuntary movement of the eye muscles [1]. Consequently, the presence of such noise may impact the accuracy and reliability of the obtained ERG signals, influencing the model's performance. 

Study protocol considerations: The employment of specific ERG protocols, namely Maximum 2.0 ERG Response, Scotopic 2.0 ERG Response, and Photopic 2.0 ERG Response, within this investigation, ensures the acquisition of ERG recordings with optimal quality on the employed equipment [2]. However, it is crucial to acknowledge that altering the study protocol, such as modifying the brightness or timing of light stimuli, could yield varying results when utilizing the model. Changes in these protocol parameters may introduce variations in the recorded ERG signals, potentially impacting the model's predictive performance and its ability to generalize to different experimental conditions or protocols. 

Dataset limitations: The dataset used in this study comprises data from both healthy subjects and subjects with retinal dystrophy [3]. While the inclusion of healthy subjects provides a baseline for comparison, the focus of the model's training and evaluation is on the detection and diagnosis of retinal dystrophy. Therefore, it is important to note that this specific model is designed and optimized for the diagnosis of retinal dystrophy and may not be applicable for the diagnosis of other diseases or conditions. The model's performance and generalizability to other diseases should be assessed separately using appropriate datasets and evaluation protocols. 

Neural network feature limitations: While neural networks have shown improved performance with larger datasets, it is essential to consider the limitations associated with the available data. The accuracy of the chosen neural networks, especially the Transformer, increases with an increase in the training subset. However, the available datasets are limited, and the different nature of the origin of the signals makes it difficult to combine such datasets. 

[1] Constable, P.; Bach, M.; Frishman, L.; Jeffrey, B.; Robson, A. ISCEV Standard for clinical electro-oculography (2017 update). Documenta ophthalmologica. Advances in ophthalmology 2017, 134. https://doi.org/10.1007/s10633-017-9573-2. 

[2] Zhdanov, A.E.; Borisov, V.I.; Dolganov, A.Y.; Lucian, E.; Bao, X.; Kazaijkin, V.N. OculusGraphy: Norms for Electroretinogram Signals. In Proceedings of the 2021 IEEE 22nd International Conference of Young Professionals in Electron Devices and Materials (EDM), 2021, pp. 399–402. https://doi.org/10.1109/EDM52169.2021.9507597. 

[3] Zhdanov, A.E.; Borisov, V.I.; Lucian, E.; Kazaijkin, V.N.; Bao, X.; Ponomarev, V.O.; Dolganov, A.Y.; Lizunov, A.V. OculusGraphy: Description of Electroretinograms Database. In Proceedings of the 2021 Third International Conference Neurotechnologies and Neurointerfaces (CNN), 2021, pp. 132–135. https://doi.org/10.1109/CNN53494.2021.9580221. 

  1. Thank you for the comment! We also added the ways how to overcome the limitations and some prospective ways for the future work:

To overcome equipment limitations in ERG recordings, the use of Erg-Jet electrodes can be considered [1]. These electrodes help minimize noise caused by the movement of eye muscles, thus improving the quality of the recorded signals. Additionally, standardizing the study protocol across different equipment setups can help address the issue of using different equipment [2].  

Regarding study protocol limitations, the adoption of standardized protocols ensures a uniform approach to ERG recordings. Furthermore, presenting detailed statistical data about the dataset used in the study helps address study protocol limitations by providing transparency and enabling researchers to evaluate the robustness and generalizability of the findings. 

To mitigate dataset limitations, it is crucial to expand the ERG dataset and include a broader range of retinal diseases. By increasing the size and diversity of the dataset, researchers can enhance the representativeness of the data and improve the model's ability to generalize to various clinical scenarios. Incorporating additional retinal diseases beyond the scope of the current study would enable a more comprehensive understanding of the model's performance and its applicability to a wider range of clinical conditions. Expanding the dataset can also help identify potential subgroups or rare conditions that may have specific ERG characteristics, contributing to the advancement of knowledge in the field. A promising development direction is the generation of synthetic ERG signals: combining the mathematical model of the signal and generative adversarial networks [3]. This will increase the training subset with signals of a similar origin. 

[1] Lu, Z.; Zhou, M.; Guo, T.; Liang, J.; Wu, W.; Gao, Q.; Li, L.; Li, H.; Chai, X. An in-silico analysis of retinal electric field distribution induced by different electrode design of trans-corneal electrical stimulation. Journal of Neural Engineering 2022, 19, 055004. https://doi.org/10.1088/1741-2552/ac8e32. 

[2] Zhdanov, A.E.; Borisov, V.I.; Dolganov, A.Y.; Lucian, E.; Bao, X.; Kazaijkin, V.N. OculusGraphy: Norms for Electroretinogram  Signals. In Proceedings of the 2021 IEEE 22nd International Conference of Young Professionals in Electron Devices and Materials  (EDM), 2021, pp. 399–402. https://doi.org/10.1109/EDM52169.2021.9507597. 

[3] Goodfellow, I.; Pouget-Abadie, J.; Mirza, M.; Xu, B.; Warde-Farley, D.; Ozair, S.; Courville, A.; Bengio, Y. Generative adversarial  nets. In Proceedings of the Advances in neural information processing systems, 2014, pp. 2672–2680. 

Round 2

Reviewer 3 Report

The authors have taken my comments into account.

I have no other comments.

The article can be accepted.

I have no significant comments.